# The genomic landscape of reference genomes of cultivated human gut bacteria

Xiaoqian Lin[1,2,10], Tongyuan Hu[1,10], Jianwei Chen[1,3,4], Hewei Liang[1], Jianwei Zhou[3], Zhinan Wu[1,5], Chen Ye[1], Xin Jin[1], Xun Xu[1], Wenwei Zhang[1], Xiaohuan Jing[6], Tao Yang[6], Jian Wang[1,7], Huanming Yang[1,7], Karsten Kristiansen[1,3,4,8] ✉, Liang Xiao[1,3,9] ✉ & Yuanqiang Zou[1,3,4,9] ✉

Culture-independent metagenomic studies have revolutionized our understanding of the gut microbiota. However, the lack of full genomes from cultured species is still a limitation for in-depth studies of the gut microbiota. Here we present a substantially expanded version of our Cultivated Genome Reference (CGR), termed CGR2, providing 3324 high-quality draft genomes from isolates selected from a large-scale cultivation of bacterial isolates from fecal samples of healthy Chinese individuals. The CGR2 classifies 527 species (179 previously unidentified species) from 8 phyla, and uncovers a genomic and functional diversity of *Collinsella aerofaciens*. The CGR2 genomes match 126 metagenome-assembled genomes without cultured representatives in the Unified Human Gastrointestinal Genome (UHGG) collection and harbor 3767 unidentified secondary metabolite biosynthetic gene clusters, providing a source of natural compounds with pharmaceutical potentials. We uncover accurate phage–bacterium linkages providing information on the evolutionary characteristics of interaction between bacteriophages and bacteria at the strain level.

Accumulating evidence has emphasized the key role of the gut microbiota in human health and disease[1,2]. Over the past few decades, associations between the composition of the gut microbiota and complex metabolic traits and diseases have been well documented[3], and fecal transplantation has been shown to hold promises for therapeutic interventions by remodeling of the gut microbiota[4]. Eventually, however, transplantation with designed well-characterized bacterial communities is desirable for clinical interventions.

Culture-independent methods provide an opportunity for the discovery of uncultivated organisms in the gut microbiota, expanding our knowledge of the composition and functional potential of gut

bacterial species. The Unified Human Gastrointestinal Genome (UHGG) collection has delivered unprecedented numbers of bacterial genomes, providing information on 4644 prokaryotic species included in the human gut. However, more than 70% of the species in UHGG lack cultivated isolates[5]. Furthermore, the existing limitations associated with metagenome-assembled genomes (MAGs), such as incomplete genomes and chimeric contigs, affect the accuracy of high-resolution taxonomic and functional inferences[6].

Cultivation-dependent studies have continued to provide new perspectives on the biology of human gut bacterial communities[7,8]. We previously presented a reference catalog of genomes of cultivated

[1]BGI-Shenzhen, Shenzhen 518083, China. [2]School of Biology and Biological Engineering, South China University of Technology, Guangzhou 510006, China. [3]Qingdao-Europe Advanced Institute for Life Sciences, BGI-Shenzhen, Qingdao 266555, China. [4]Laboratory of Genomics and Molecular Biomedicine, Department of Biology, University of Copenhagen, Universitetsparken 13, 2100 Copenhagen, Denmark. [5]College of Life Sciences, University of Chinese Academy of Sciences, Beijing 100049, China. [6]China National GeneBank, BGI-Shenzhen, Shenzhen 518120, China. [7]James D. Watson Institute of Genome Sciences, Hangzhou 310058, China. [8]PREDICT, Center for Molecular Prediction of Inflammatory Bowel Disease, Faculty of Medicine, Aalborg University, 2450 Copenhagen, Denmark. [9]Shenzhen Engineering Laboratory of Detection and Intervention of Human Intestinal Microbiome, BGI-Shenzhen, Shenzhen, China. [10]These authors contributed equally: Xiaoqian Lin, Tongyuan Hu. ✉e-mail: kk@bio.ku.dk; xiaoliang@genomics.cn; zouyuanqiang@genomics.cn

human gut bacteria (CGR) improving taxonomic annotation and functional inferences[9]. Cultivated gut microbial resources enable better bioprospecting of the gut microbiota including identification and isolation of carbohydrate-binding enzymes, bioactive molecules, bacteriophages, and next-generation probiotics.

Here we present a substantially expanded Cultivated Genome Reference, termed CGR2, of the human gut microbiota. The CGR2 comprises 179 previously unidentified species with high-quality genomes. We provide information on carbohydrate-active enzymes and secondary metabolite biosynthetic gene clusters of the gut microbiota, assigning specific taxa to functions, and accurate linkages between viruses and microbial hosts. Finally, we conducted a genome-wide analysis of a representative species. We envisage that this collection of bacterial genomes will constitute a valuable source for future studies on the gut microbiota furthering in-depth knowledge of the gut microbiota.

## Results

### The expanded repertoire of isolate genomes in CGR2

In continuation of our previous work on cultivation and sequencing gut-resident microbes, a total of ~20,000 bacterial isolates were cultivated. 4066 of these isolates were selected for whole-genome sequencing generating 3324 high-quality genomes with more than 90% completeness and less than 10% contamination (Supplementary Fig. 1, and Supplementary Data 1). We subsequently clustered the 3324 genomes into 527 species-level clusters on the basis of 95% average nucleotide identity (ANI), of which 189 clusters (1804 genomes) were not included in the CGR. The clusters were distributed between 8 phyla, of which Bacillota represented more than half of the clusters (1805 genomes, 343 clusters). Notably, Synergistota, Thermodesulfobacteriota, and Verrucomicrobiota were newly included in CGR2 compared to CGR (Fig. 1a). Of the 527 species-level clusters, 179 were not classified at the species-level, and 21 lacked a genus-level match (Supplementary Data 2), indicating that these clusters harbor previously unidentified species. Whereas some important species were clearly underrepresented in CGR, CGR2 encompasses a large number of high-quality genomes of *Bifidobacterium longum*, *Bifidobacterium pseudocatenulatum*, *Bifidobacterium adolescentis*, *Escherichia coli*, and *Enterococcus faecalis*, which can be used for species pan-genome and diversity analyses (Supplementary Fig. 2).

The distribution of 527 species-level clusters in 3 representative metagenomic cohorts of different origins, including China (a part of 4D-SZ)[10], the Netherlands[11], and HMP (Human Microbiome Project) is shown in Fig. 1b. The prevalence of *Flavonifractor plautii*, *Bacteroides uniformis*, and *Bacteroides caccae* exceeded 95% in three cohorts (Supplementary Data 3). Beta diversity showed that there were significant differences between the 527 clusters in the three cohorts ($R^2 = 0.2984$, $P < 0.001$), especially between the cohorts from China

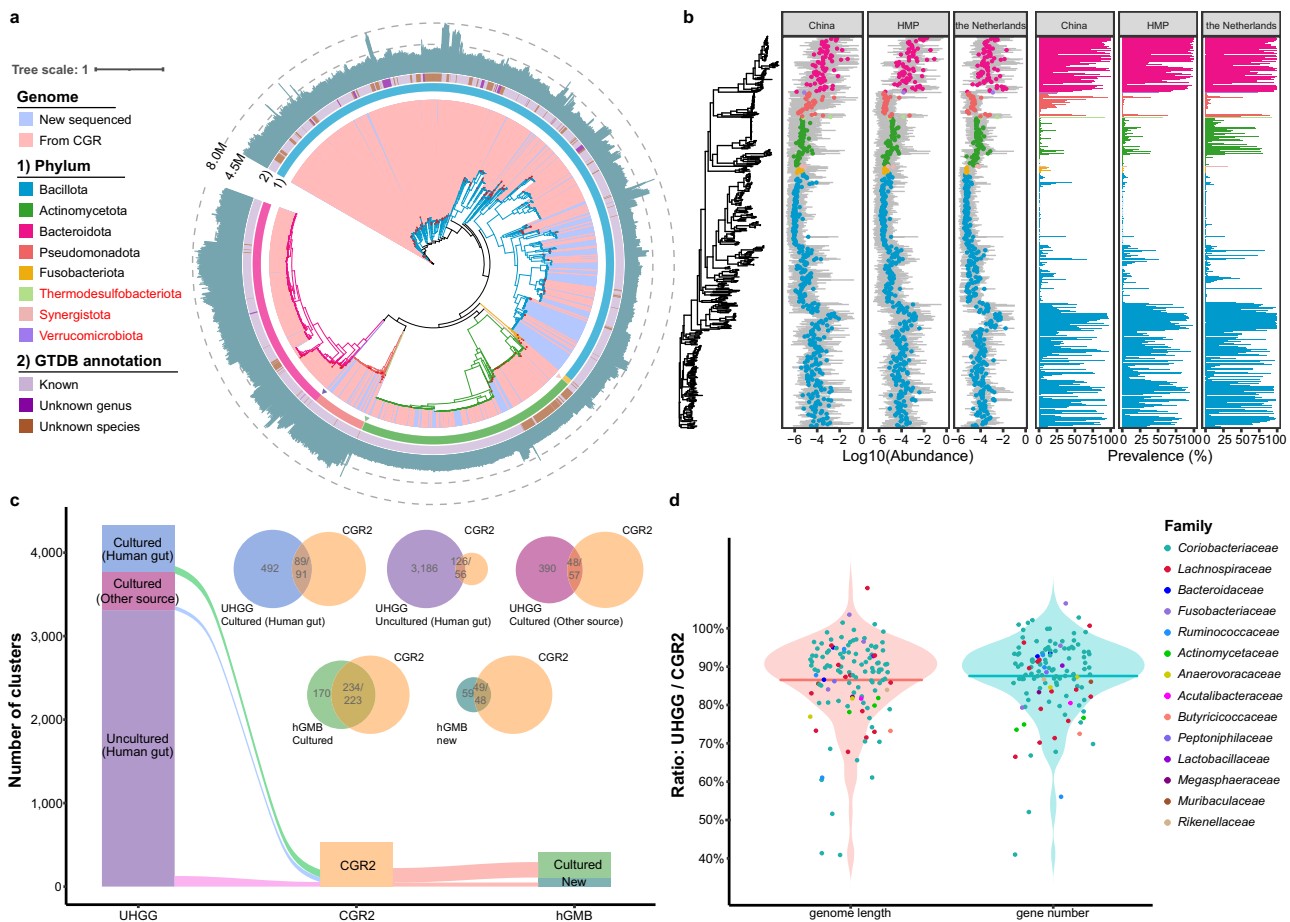

**Fig. 1 | Taxonomic profile of CGR2. a** Phylogenetic analysis of 3324 genomes. Color range indicates the 1804 newly sequenced genomes (blue) and the 1520 CGR genomes (pink). Singleton genomes are marked with red dots at the end of the clade. The first layer depicts the GTDB phylum annotation, the second layer describes the matching to the GTDB database at the species and genus level, and the circumferential bar plot (dark blue) illustrates the genome size. **b** Abundance and prevalence of 527 representative clusters in healthy cohorts of China, HMP, and the Netherlands. Gray box, Log10 (relative abundance); Dot, median of log10 (relative abundance); Bar, prevalence; Color, phylum. **c** Matching of CGR2 to the hGMB and UHGG genome collections. The Venn diagrams are colored according to the origin of the samples and the numbers are indicated. **d** The ratio of the genome length (median: 88.84%) and gene number (median: 89.33%) of the UHGG-Uncultured relative to CGR2 in the mapped genomes of each family. A dot represents a UHGG genome, and different colored dots indicate different family.

and the Netherlands (Supplementary Fig. 3a). The correlations between the 527 clusters and the coordinates of microbial communities suggested that *Prevotella* sp. (Cluster 62), *Bacillus luti*, *Paenibacillus* sp. (Cluster 281), and *Paenibacillus macerans* had the most significant impact on the distribution of these clusters ($P < 0.001$, Supplementary Fig. 3a, and Supplementary Data 3). Species of *Prevotella* are important members of the human gut microbiota, and recent studies suggested a reclassification of *Prevotella* into seven genera[12,13], which display different metabolic characteristics. In addition, compared with the cohort from the Netherlands, the medians and means of the 527 clusters in the cohort from China and the HMP cohort were more similar (Supplementary Fig. 3b, c). Examining the distribution of the 179 previously unidentified species in the different populations, we found that the average abundance of these previously unidentified species in the Chinese population was 0.08%, which was significantly higher than that in the other two cohorts ($P < 0.0001$, Supplementary Fig. 4a). However, the occurrence is much lower than that in the cohort from the Netherlands ($P < 0.001$, Supplementary Fig. 4b). 42 species were significantly enriched in the Chinese cohort (Supplementary Fig. 4d). Of note, the inclusion of previously unidentified species in CGR2 significantly improves metagenomic reads mapping rate in Chinese and non-Chinese populations, especially for the cohort from the Netherlands ($P < 0.0001$, Supplementary Fig. 4c, and Supplementary Data 3).

Mapping the CGR2 genomes against the 3312 genomes representing uncultured species and 438 genomes of cultured species from other sources, we found that 146 CGR2 genomes matched 126 UHGG genomes of uncultured species, and that 136 CGR2 genomes matched 48 genomes of cultured species from other sources in UHGG, illustrating how our collection increases the taxonomic diversity of cultivable microorganisms in the human gut (Fig. 1c). Comparing the matches from 146 CGR2 genomes and 126 UHGG-uncultured genomes, we found not surprisingly that the gene number and scaffold N50 of the genomes obtained by sequencing of cultivated isolates were significantly higher than those from MAGs ($P < 0.0001$, Supplementary Fig. 5). We compared genome length and gene differences of the genome sets to explore the assembly gaps between genomes based on isolates and MAGs. In general, less than 90% of the genome length and gene number of culture-based genomes were covered by the corresponding MAGs (Fig. 1d). Comparing genomes based on isolates with the corresponding MAGs, we identified 1543 unique genes present in the isolate-based genomes, but absent in the MAGs with *Erm* being the gene most frequently missing in the MAGs (Supplementary Fig. 6, and Supplementary Data 4). Only 286 CGR2 isolated genomes (91 clusters) mapped to 89 human gut culture-based genomes originating from non-Chinese samples in the UHGG, possibly reflecting the differences in the composition of the gut microbiota in individuals of different ethnicity and/or living in different geographical locations[14]. The Broad Institute-OpenBiome Microbiome Library (BIO-ML)[15] is a human gut strain collection established from FMT donors. A comparison revealed that 82 of the BIO-ML species-level clusters were also included in CGR2, and only 22 BIO-ML species-level clusters were not included in CGR2, implying an 84.44% taxonomic novelty in CGR2 (Supplementary Fig. 7a). We also compared the genomes of CGR2 with hGMB[16], a recent cultured genome collection based on Chinese samples, showing that 2306 of the CGR2 genomes covered 57.92% of the hGMB genomes, including 49 of 108 newly characterized and classified species. Overall, 144 clusters from CGR do not exist in any existing collection, and the newly sequenced genomes of CGR2 contributes with 45 unique clusters (Supplementary Fig. 7b). Further, we found that 89 of 179 previously unidentified species were not represented in UHGG, BIO-ML or hGMB, including one cluster being annotated only at the class-level (Supplementary Fig. 8a, and Supplementary Data 2). It is worth noting that these underrepresented previously unidentified species may be widely distributed in the cohorts from China and the Netherlands, and

in the HMP cohort (Supplementary Fig. 8b–d). In addition, there are still 31 previously unidentified species in CGR2 that are only represented by MAGs, while we provide the cultured strains to facilitate subsequent taxonomic characterization (Supplementary Fig. 8a).

## The distribution of carbohydrate-active enzymes (CAZymes)

To examine the function of the culture isolates of CGR2, we performed a comprehensive in-depth analysis of carbohydrate-active enzymes. Notably, isolates of the Bacteroidota phylum harbored the largest and most diverse CAZyme repertoires (Fig. 2a, and Supplementary Fig. 9a), consistent with previous studies[17,18]. 86.08% of the predicted CAZyme genes in CGR2 belong to the glycoside hydrolases family (GH) and the glycosyltransferases family (GT) (Supplementary Data 5). We next explored the potential for utilization of dietary fibers (including pectin, cellulose, and inulin) in the 3324 genomes. Bacteroidota contained more GH or polysaccharide lyase (PL) genes reported to potentially be involved in the degradation of dietary fiber (Supplementary Fig. 9b). Bacteria belonging to the Bacteroidota phylum are able to utilize a broad range of carbohydrate substrates[19], and they may play a role in initiating the primary breakdown of dietary polysaccharides in the human gut.

We next collected all enzymes involved in decomposition pathways of pectin, cellulose and inulin, and synthetic pathways of the three short chain fatty acids (SCFAs), acetate, propionate, and butyrate, from the Kyoto Encyclopedia of Genes and Genomes (KEGG) database to screen for potential glycan-degrading and SCFA-producing strains (Supplementary Fig. 10). The result showed that most strains in CGR2 had potentials for breaking down inulin (96%) and producing SCFAs, acetate (100%), propionate (97%), and butyrate (79%) (Supplementary Data 6). 193 Strains belonging to 42 genera were predicted to possess at least one complete enzymatic pathway for each of these metabolic pathways (Fig. 2b). Ten genera, *Bacteroides*, *Bifidobacterium*, *Enterococcus*, *Phocaeicola*, *Blautia*, *Collinsella*, *Streptococcus*, *Enterocloster*, *Escherichia*, and *Mediterraneibacter* contained the majority of strains with complete pathways for decomposing glycans or synthesizing SCFAs.

Human milk oligosaccharides (HMOs) play important roles in the early nutrition of beast-fed infants and serve as substrates supporting the growth of important members of the *Bifidobacterium* genus dominating the gut in early life[20]. Large type I and II HMOs, two main types of HMOs, can be broken down by GH95 AfcA, GH29 AfcB, and GH33 SiaBb releasing lacto-N-tetraose (LNT) and lacto-N-neotetraose (LNnT)[21,22] (Fig. 2c). GH136 (lacto-N-biosidase) and GH112 (lacto-N-biose phosphorylase) are the core hydrolases in type I HMO degradation, and GH2/42 (β-galactosidase) and GH20 (β-N-acetylhexosaminidase) are the core hydrolases in type II HMO degradation[21–23]. The two carbohydrate-binding module families, CBM32 and CBM51, are possibly relevant to HMO degradation[21]. We next explored the distribution of these CAZyme families in our genome collection. In CGR2, GH2 is the most widely distributed GH, present in 81.6% of the genomes. Notably, 337 and 1546 genomes contained complete CAZyme families for degradation of type I and II HMOs, respectively (Supplementary Data 7). Members of the phylum Bacteroidota possess GH2, GH20, GH29, GH95, and CBM32, but lack GH112 and GH136, the key CAZyme families involved in type I HMO degradation (Fig. 2c). In addition to the presence and absence of CAZyme families involved in HMO degradation, unconstrained principal coordinate analysis (PCoA) was used to explore the similarity of the gene numbers of these CAZyme families. This analysis showed that members of Bacteroidota formed a distinct central cluster (Fig. 2d).

For bifidobacteria, GH2 and GH42 are the two most prevalent GH families, included in all genomes of this genus (Supplementary Fig. 11a). *B. bifidum* and *B. longum* contain type I and II HMO degrading CAZyme families. In addition, *B. bifidum* harbors more hydrolases and CBMs belonging to CAZyme families involved in HMO degradation, suggesting that this species is highly adapted for use of HMOs[21]. In

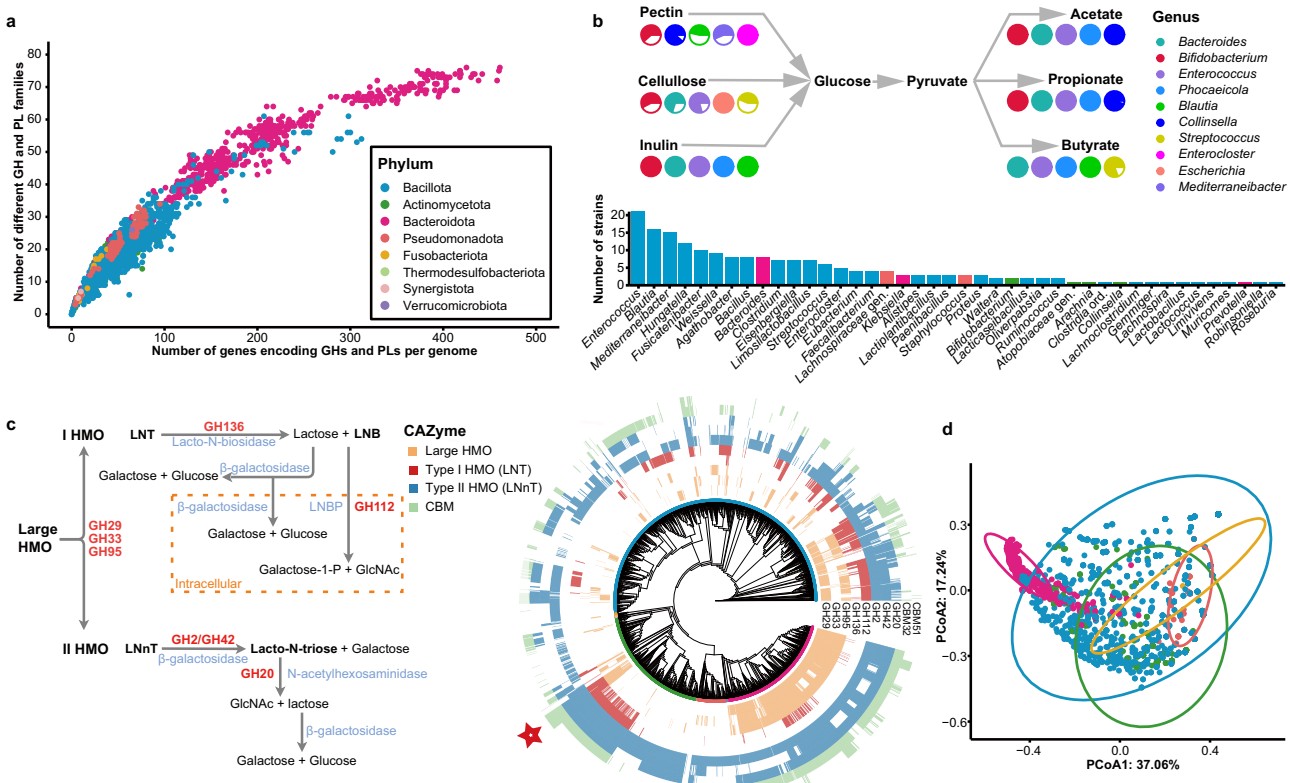

**Fig. 2 | Distribution of CAZymes in the CGR2 genomes. a** Differences in the numbers of genes encoding GH and PL, and the numbers of GH and PL families represented in these genomes. **b** Simplified illustration showing the degradation of dietary fibers and the synthesis of SCFAs by the dominant genera (see "Methods"). Ratio of the numbers of strains in top five genera with complete pathways and in genera of CGR2 is shown as pie plot charts (see Supplementary Data 6). Phylum annotation of bar chart as in (**a**). **c** Simplified schematics of pathways involved in HMO degradation and phylogenetic tree of 3324 genomes combined with the heatmap on the outermost layer indicating the presence or absence of HMO-degrading GH and CBM genes. The colored rings around the tree represent the taxonomic classifications using the same annotation as in panel (**a**). LNT, lacto-N-tetrose. LNB, lacto-N-biose. LNBP, lacto-N-biose phosphorylases. LNnT, lacto-N-neotetraose. Star marks bifidobacteria. **d** PCoA based on Bray–Curtis dissimilarity of the numbers of HMO-degrading CAZyme genes. Ellipses cover 95% of the genomes for each phylum. Phylum annotation as in (**a**).

addition, our analyses demonstrated that *Roseburia*, a butyrate producer possessing multiple HMO utilization CAZyme families, also has a potential for metabolizing HMOs consistent with a previous study[23] (Supplementary Fig. 11b). These analyses support the notion that Bacteroidota together with bifidobacteria, belonging to the Actinomycetota, in CGR2, play a key role in promoting the release of HMO central moieties, consistent with previous reports[21,24].

## Identification of genes involved in the synthesis of secondary metabolites in the gut microbiome

We performed a comprehensive analysis of secondary metabolite biosynthetic gene clusters (SMBGs) using anti-SMASH (v4.2.0)[25]. Notably, 4132 gene clusters involved in the generation of secondary metabolites were identified in 2049 genomes (Supplementary Data 8 and "Methods"). Of these gene clusters, the most abundant were inferred to participate in the biosynthesis of sactipeptides (907), followed by non-ribosomal peptide synthetases (NRPSs, 804) and bacteriocin (740). A total of 24 different biosynthetic types were predicted to be present in the 8 phyla present in CGR2, with Bacillota harboring the highest abundance of SMBGs and a broad distribution of specialized metabolites (Fig. 3a).

4132 SMBGs clustered into 7 classes (978 gene cluster families, GCFs). Only 46 GCFs are included in the MIBiG database reference with known functions, indicating that most of the SMBGs we predicted lack experimental verification and might potentially represent novel functions (Supplementary Fig. 12). The largest class is ribosomally produced and post-translationally modified peptides (RiPPs), which include bacteriocins, lantipeptides, and sactipeptides. To better understand the diversity of SMBGs of relevance for RiPPs, sequence similarity networks were constructed for 2303 SMBGs (528 GCFs). The networks showed that most predicted RiPPs were from the Bacillota phylum, indicating that Bacillota could be a potentially abundant source of RiPPs (Fig. 3b).

Ruminococcin A (RumA), naturally produced by the strictly anaerobic bacterium *Ruminococcus gnavus* E1, has high activity against pathogenic *Clostridium* spp, and has been used for clinical treatment[26]. However, due to low production yields (<1 μg/L) and difficult cultivation of *R. gnavus* E1, high-quality production of RumA is challenging[27]. Our results showed that 7 SMBGs mined from *Faecalimonas umbilicata*, *Mediterraneibacter faecis*, *Fusicatenibacter saccharivorans*, *Blautia* sp., and *Waltera intestinalis* harbor genes related to the biosynthesis of RumA, suggesting that these bacteria may be used as a potential alternative source for the production of RumA (Fig. 3c). NRPSs constituted the second largest class, containing 816 SMBGs (198 GCFs), mainly from the Bacillota and Pseudomonadota phyla (Supplementary Fig. 13a). By analyzing 8 clans containing SMBGs with reliable experimental evidence, we discovered that one of the clans is related to dipeptide aldehydes, highly potent cell-permeable protease inhibitors, initially detected in *Ruminococcus* sp[28], while the SMBGs of this clan were all derived from *Blautia* (Supplementary Fig. 13b). Similarly, for the other 5 classes of SMBGs, we also conducted a network similarity analysis (Supplementary Fig. 14). These results revealed an unexplored novelty and diversity of SMBGs from human gut microbes.

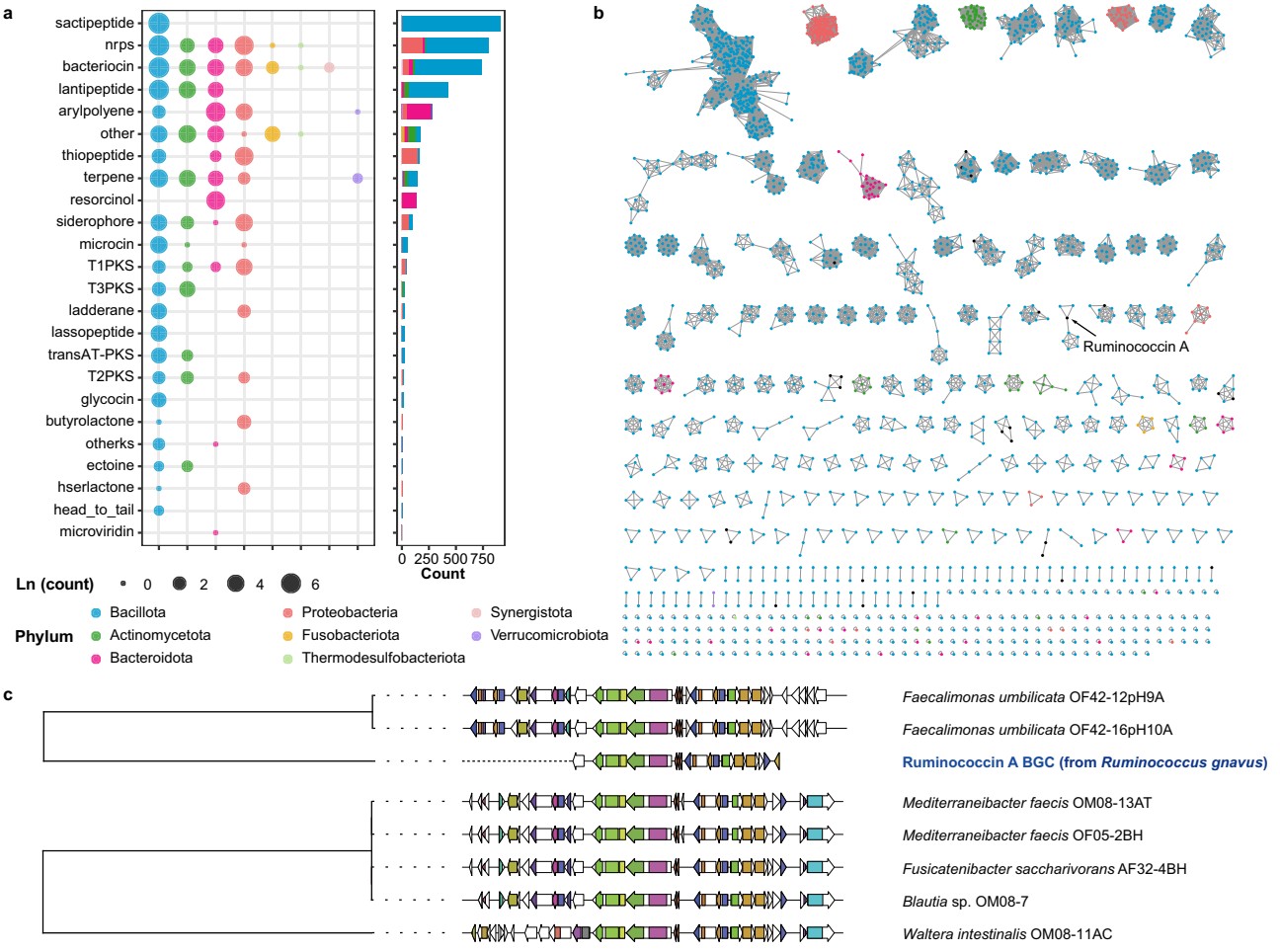

**Fig. 3 | Abundant SMBGs in CGR2. a** The distribution of SMGBs in different phyla. The size of the dot indicates the number of SMBGs. **b** Sequence similarity network of RiPPs. Nodes represent sequences of BGC domains and are colored by the phylum of the BGC-derived genome (black nodes: reference BGC). Edges drawn between the nodes correspond to pairwise distances. **c** The sequence evolution relationship of the RumA clan; different shapes indicate different modules.

## Prediction of prophages in the isolated genomes and phage-bacteria interactions in the gut microbiota

To construct detailed networks between phages and the host bacteria, we identified bacteriophages in the culture-derived genomes of CGR2 using VirSorter[29]. A total of 14,249 potential viral sequences were predicted from 3324 genomes, with 6274 being considered as "most confident" and "likely" phages and prophages[30]. After quality evaluation by CheckV, 22 phage genomes were assigned as complete, 150 as high-quality, and 2648 as medium-quality, representing the Viruses of the Cultivated Genome Reference (termed CGRv) (Supplementary Fig. 15a, and Supplementary Data 9). Comparing with the Gut Phage Database (GPD)[31] and Metagenomic Gut Virus (MGV)[32], 60.53% of the phages in CGRv were not reported previously (Supplementary Fig. 15b). In addition, we discovered three jumbo phages (>200 kb of length), a group of phages rarely described previously[33].

A phylogenetic tree was constructed to explore the evolutionary characteristics of phages in the gut microbiota. Notably, phages from Actinomycetota clustered within a single clade (Fig. 4a). However, phages from Bacillota were widely distributed throughout the phylogenetic tree, reflecting a high level of variation of phages in this phylum. To investigate the phage diversity of the CGRv, we conducted a clustering and taxonomic approach using VConTACT v2.0[34] (Supplementary Data 10). In total, 2117 clustered phages were divided into 317 virus clusters (VCs), hosted by 315 bacterial species (59.8% in CGR2). For the taxonomy of phages in CGRv, only 269 phage genomes (12.7%) could be assigned to known families, including *Siphoviridae* (135),

*Myoviridae* (108), and *Podoviridae* (26), whereas the vast majority of the phages were previously unidentified at the family level.

Next, we determined the host range of phages and phage-host network visualized using Cytoscape[35] (Supplementary Fig. 16a). VC_399, a most infectious VC of CGRv, has the broadest range of bacterial hosts and can infect 31 bacterial species from Bacillota, Actinomycetota, and Synergistota (Supplementary Fig. 16b). In addition, *Clostridium fessum* and *Phocaeicola vulgatus* were the most targeted species that may be infected by 15 different VCs (Supplementary Fig. 16c). Strikingly, four VCs were contained in the genomes of bacteria spanning different phyla, including VC_399, VC_67, VC_195, and VC_220 (Fig. 4b). This may represent horizontal gene transfer events between host bacteria in different phyla. For the quantification of the host range of CGRv, we found that more than half of the VCs (167/317) may infect multiple species of bacteria (Supplementary Fig. 16d).

We conducted an identification of proteins encoded by CGRv. A total of 212,369 proteins were predicted and clustered as 25,345 non-redundant protein clusters (Supplementary Data 11). 169,859 proteins (79.98%) were uncharacterized, while 7394 proteins (3.48%) were annotated as related to phage structure. Particularly, PC_000663, a protein cluster involved in O-antigen conversion and bactoprenol glucosyltransferase, which are considered to be related to polymyxin resistance, was predicted to be encoded by 34 phages with 34 different bacterial hosts in the Bacillota and Pseudomonadota phyla (Fig. 4c). The proteomic tree of PC_000663 showed that *Lactococcus garvieae* TM115-50 and *Lactococcus petaurid* TM115-81 phylogenetically formed the

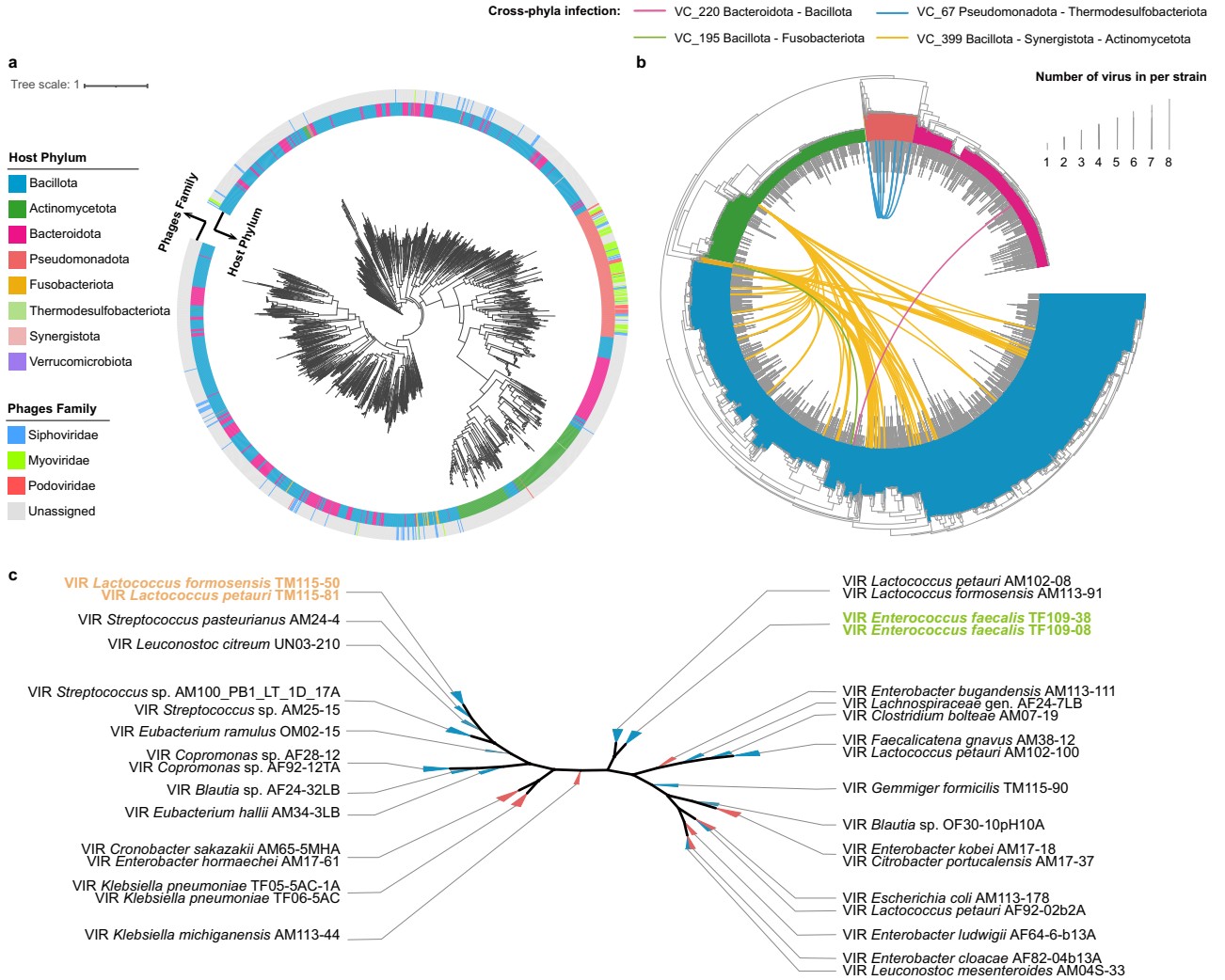

**Fig. 4 | Host range of phages and interactions between phages and bacteria.**
**a** Phylogenetic tree of 1919 phage sequences. The inner circle is colored according to the host phylum of the phages, and the outer circle is colored according to the phage family (227 phages were assigned). **b** Phylogenetic tree of CGR2 genomes showing phage host range. The height of the gray bars denotes the number of viruses the host harbors. Connections in four colors where each color represents one VC that can infect hosts in different phyla. **c** Phylogenetic analyses of PC_000663. Phages are named "VIR_host species_host number". Leaf colors indicate to which phylum phages belong.

closest clade. We found that the two species were infected by different VCs that can encode PC_000663, suggesting that the homology within PC_000663 may reflect horizontal gene transfer events.

## Genome-wide analysis of *Collinsella aerofaciens* reveals high genomic and functional variations

Many bacteria exhibit wide variations between different niches. In this study, we discovered more than 10 clusters of *Collinsella aerofaciens* in CGR2, indicating a high level of genomic diversity of these bacteria. The genomic ANI clusters showed that 130 isolated genomes from CGR2 and 67 genomes retrieved from NCBI (including 10 UHGG isolated strains) were divided into 19 clades, included 8 singleton clades (Fig. 5a). Interestingly, the distribution of the 19 clades was inconsistent with their phylogenetic clades in the SNP phylogenetic tree. We then clustered these 197 genomes into 5 groups using CDS sequences highly consistent with the SNP phylogenetic tree, of which group 2 and group 4 were singletons (Fig. 5b), suggesting that the mutation in non-protein-coding intergenic regions is one cause of high genomic diversity. SNPs analysis showed that less than 10% SNPs were present in the intergenic regions, but more than twice the number of insertions or deletions (InDels) were detected in the

intergenic regions compared to CDS regions (Fig. 5c, d, and Supplementary Fig. 17a, b), indicating that these species had rapid InDel structural variations in the intergenic regions. We checked the 10 CDSs with top mutation frequency and found that the variant type and frequency appeared to correlate with cluster and country (Supplementary Fig. 17c). Furthermore, the gene copy number of several CAZy enzymes differed significantly between different groups, with the gene numbers of GH1, GH4, and CE9 in the group 3 strains being higher than those of group 1 and group 5 strains ($P < 0.01$, Supplementary Fig. 18), which may underlie the differences in the capacity for utilization of cellulose[36], cleavage of the glycosidic bond[37], and biosynthesis of amino-sugar-nucleotides[38]. Taken together, we greatly increased the reported genomic diversity of *C. aerofaciens* providing important insights for uncovering the genomic and functional differences of different groups of *C. aerofaciens*.

## Discussion
In this study, we used a large-scale culture-based method to obtain human gut microbial genomes, expanding the collection of the Cultivated Genome Reference (CGR2) released as a valuable resource for more comprehensive exploration of human gut microbes. CGR2

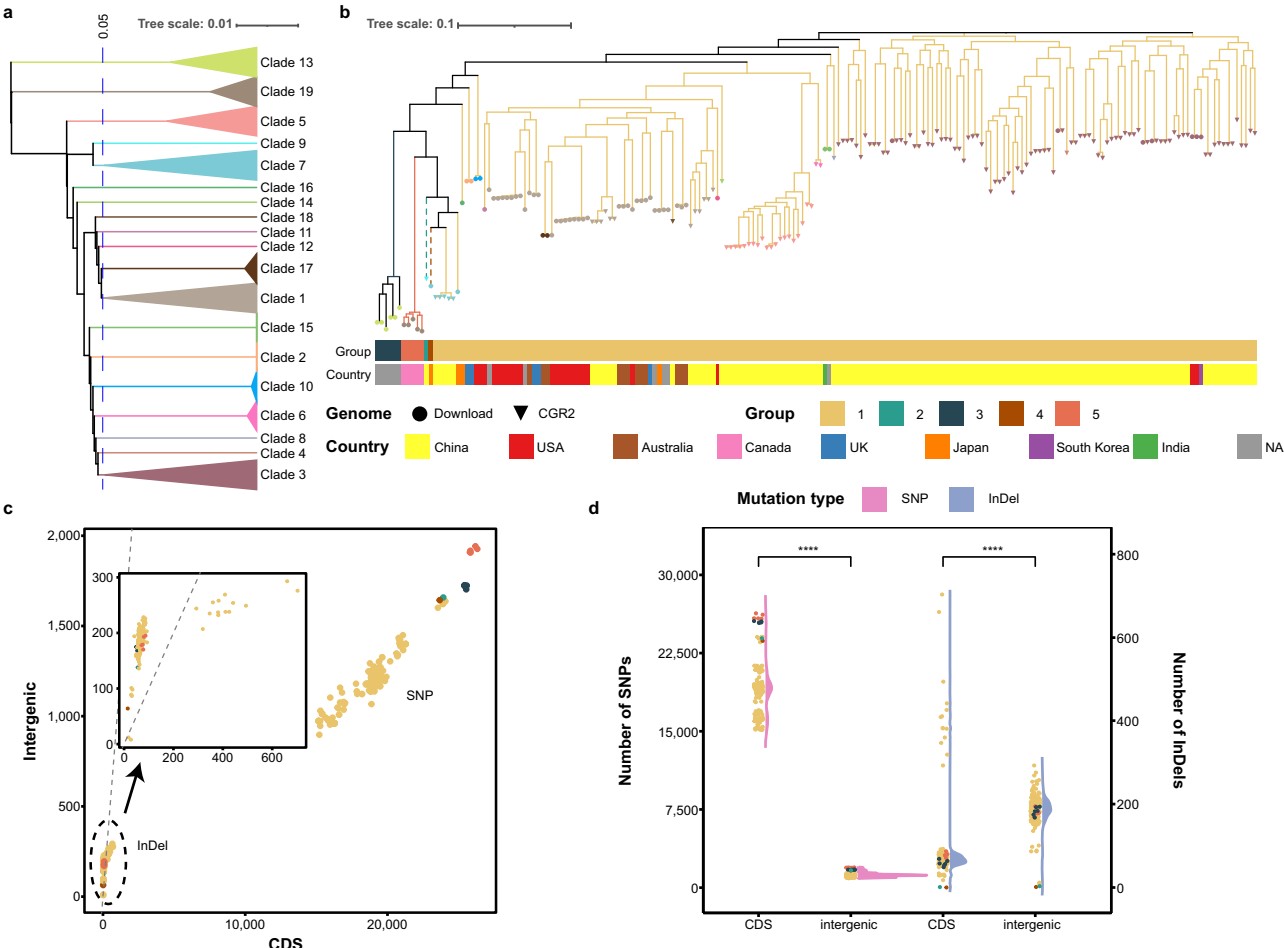

**Fig. 5 | Genome-wide analysis of 197 *Collinsella aerofaciens* genomes. a** Whole-genome phylogenetic tree constructed based on ANI. According to the distance of 0.05, the genome can be divided into 19 clades. **b** Consistency between the genomic SNP phylogenetic tree and the 5 groups of CDS sequences. Tip shapes and colors represent the genome source and ANI clade, respectively. Dashed lines indicate the singletons in group 2 and group 4. For each genome, the first layer represents the group, the second layer denotes the country where the strain was isolated. **c** Distribution of InDels and SNPs in CDS regions and intergenic regions for each genome. Colors represent the group. **d** The InDel and SNP variation statistics in CDS regions and intergenic regions. Colors represent the group. *P* values are from Wilcoxon rank-sum test (two-sided) (****$P < 0.0001$).

presents 3324 high-quality genomes from 527 species, including 179 previously unidentified species. However, laboratory-based phenotypic analysis, taxonomic naming and species description of these 179 previously unidentified species are still limited. Correct taxonomy of previously unidentified species is clearly warranted, and therefore, polyphasic taxonomy studies of these previously unidentified species will be conducted in our future studies. When aligned with metagenomic samples from healthy individuals in cohorts from China and the Netherlands, and the HMP cohort, we found that 527 species were widely distributed in the Chinese cohort and the HMP cohort, but less prevalent in the cohort from the Netherlands. The bacterial community structure in the Chinese cohort and the HMP cohort were more similar than that of the cohort from the Netherlands. The CGR2 genomes exhibit limited matches with the previous comprehensive collections, UHGG, BIO-ML, and hGMB, which highlights the taxonomic novelty. Recently, the recovery of genomes via cultivation-independent methods has greatly expanded the reported diversity of microorganisms[5,39]. In our collection, 40.42% of the species were represented as singletons containing only one strain, limiting the strain level diversity analysis of these species. We also report on a bias in species cultivated from samples from individuals of different ethnicity and geographical location consistent with a previous study[5], suggesting that extensive culture-based studies are indispensable for a

comprehensive understanding of the human gut microbiota globally. It is conceivable that the differences between the isolated strains and MAGs that we have found may also be caused by strain differences within the species pan-genome, especially for genomes collected from different geographic locations. Further studies are clearly warranted to elucidate to what extent different ethnicities and geographic location contribute to the observed differences.

We observed that CAZyme genes involved in degradation of dietary fibers were more abundant in members of the phylum Bacteroidota than in members of the Bacillota phylum, and that these genes presented smaller presence and absence variation (PAV). Members of the *Bifidobacterium* genus contain more core type I or II HMO degrading CAZyme genes, and notably *Bifidobacterium bifidum* possesses a complete HMO metabolism pathway. In this work, we also confirmed that *Roseburia* possesses the metabolic capabilities for metabolizing HMOs, consistent with a previous report[23]. Interestingly, Bacteroidota contains a high number of large HMO genes. Thus, Bacteroidota may possibly play a key role in the primary utilization of both dietary fibers and HMOs. The unidentified members of *Roseburia* and Bacteroidota in the gut microbiota may also be involved in this process, providing new insight into the metabolism of HMOs.

The small biologically active molecules produced by the microbiota are not only of importance for microbe-microbe and

microbe-host interactions[40], but have also wide applications as pharmaceutical, agricultural, and dietary agents. Traditionally, laboratory experiments are used to mine novel natural products based on specific strains. In recent years, genome-mining approaches have attracted more and more attention[41]. Thus, previous studies have exploited secondary metabolites of environmental microorganisms[42,43], while a few studies have focused on the secondary metabolite resources of human microorganisms[44]. The CGR2 genomes have great potential for exploration of secondary metabolites, highlighted by the existence of hundreds of SMBGs without known functions in the human gut microbiome providing an option for the discovery of natural products with biological activity in the human microbiota. With the generation of this massive genome reference, we are able to explore the interplay between SMBGs and the human host at the strain level, providing more comprehensive insight into the possible contribution of gut microbes to host health.

We performed a large-scale analysis of phages in the isolate-based genomes and constructed a comprehensive gut phage-bacteria interaction network identifying a large number of interactions between phages and host-bacteria. The interaction networks revealed a broad infection range of phages in the human gut microbiota, particularly in bacteria from the Bacillota phylum, which demonstrated that most of the phages (56.47%) in the human gut are not very specific for their bacterial host. We even identified four VCs able to infect bacteria from different phyla, adding to our understanding of the specificity of phages. Such promiscuous transduction behavior of phages may have a pronounced impact on the gut microbiota affecting the potential application of phages for combating pathogens with multidrug-resistance.

Our study also provides a comprehensive genomic resource, which in combination with published genomes from other source enables a genome-wide analysis of specific species. Thus, we demonstrate an unexpected genomic diversity of *Collinsella aerofaciens* in the human gut, including the discovery of numerous variations present in the intergenic regions of *C. aerofaciens* genomes compared with CDS regions, highlighting an underestimated genome diversity of this species. We emphasize that strain-level diversity should be taken into account for the exploration of function and evolution of specific bacterial strains.

More and more research is moving from studies of association to studies of causality and interventions. The cultured isolates enable a more detailed exploration of the function of specific bacteria in vitro and in vivo. Most importantly, a large collection of bacteria and genomes offers options to develop probiotics and secondary metabolite products for use as alternative therapeutic modalities for clinical interventions.

## Methods

### Sample collection and culturing
The sample collection was approved by the Institutional Review Board on Bioethics and Biosafety of BGI under the number BGI-IRB 20106-T2.

299 healthy human donors not taking any drugs during the last month before sampling were included for fecal collection. Candidate donors, with reference to Liu et al.[16], were considered healthy donors if they presented without any diagnosed disease, regardless of their sex. The sampling and culturing method were as described by Zou et al.[9]. The 16S rRNA gene sequence of each isolates was amplified and sequenced as previously described[45]. The taxonomy of isolates was determined and checked by blasting the 16S rRNA gene sequences against reference sequences in the EZBioCloud Server (https://www.ezbiocloud.net/)[46]. The study was conducted in accordance with the Declaration of Helsinki and informed consent was provided by all donors.

The study conforms to the "Guidance of the Ministry of Science and Technology (MOST) for the Review and Approval of Human Genetic Resources", and the public use of our data has been approved under the numbers 2022BAT2332 and 2022BAT2377.

### Genome sequencing, assembly, quality assessment, and gene prediction
All the isolates were grouped into species-level clusters based on a threshold of 98.7% identity of the 16S rRNA gene sequence[47]. Our strategy for selecting strains for WGS were (1) representation of candidates for unidentified taxa, (2) covering as many taxa as possible of the strains cultivated in the study, (3) important species from various donors. Among the 1804 newly sequenced strains, 1282 were sequenced using the MGISEQ-T7 platform, and the remaining 522 were sequenced by the Illumina Hiseq 2000 platform. The methods of whole-genome sequencing and de novo assembly were as described by Zou et al.[9]. Gene numbers were calculated using GeneMarkS-2 (v1.10)[48]. Genome quality was evaluated using CheckM (v1.1.2)[49] 'lineage_wf' workflow to select genomes with >90% completeness and <10% contamination as high-quality genomes. The gene prediction and enzyme annotation of the 3324 genomes were performed on Prokka 1.14.6[50].

### Phylogenetic and taxonomic analyses
The pairwise ANI alignment was performed for the 3324 genomes using fastANI (v1.32)[51], and hclust from the R package was used to cluster at proposed cutoff species level (ANI ≥ 95%). GTDB-Tk[52] (v2.1.0, database r207[53], 'classify_wf' function and default parameters) was used to perform taxonomic annotation of each genome and reconstruct the maximum-likelihood phylogenetic tree based on 120 conserved single-copy genes. Taxa nomenclatures have been updated based on the latest valid published name. The phylogenetic tree was viewed using the display and annotation tool iTOL (v6.1.1)[54].

### Comparison of the distribution of species-level clusters in healthy individuals in cohorts established in China, in the Netherlands, and in the HMP
Human gut metagenome sequencing data of a Chinese cohort (a part of 4D-SZ)[10] was downloaded from the CNGB Sequence Archive (CNSA)[55] (https://db.cngb.org/cnsa/) of China National GeneBank DataBase (CNGBdb)[56] under the accession code CNP0000426. Gut metagenome data of healthy individuals from a cohort established in the Netherlands[11] was retrieved from the European Genome-Phenome Archive under accession EGAS00001005027. Gut metagenome data of healthy individuals from HMP was downloaded following the link https://portal.hmpdacc.org/. The 527 species-level clusters were built as a custom genome database by Kraken v2.1.2[57] and Bracken v2.5[58]. Kraken2 and Bracken were also used to calculate the read numbers of the 527 species-level clusters in the cohorts established in China, in the Netherlands, and in the HMP. Median and mean of the relative abundances of the 527 clusters in the three cohorts were calculated, and the correlations between the medians and means of the three cohorts were analyzed based on Spearman's rank correlation coefficient. To calculate prevalence, a threshold of 0.01% relative abundance was used to define the occurrence of a cluster in one sample[59].

### Alignment with other genome collections
We downloaded 4644 representative genomes (including 894 human gut cultured genomes, 3312 uncultured genomes, and 438 other source cultured genomes) from UHGG (http://ftp.ebi.ac.uk/pub/databases/metagenomics/mgnify_genomes), 3423 isolated genomes of Broad Institute-OpenBiome Microbiome Library (BIO-ML, https://www.ncbi.nlm.nih.gov/bioproject/PRJNA544527), and 404 isolated genomes from hGMB (https://www.ncbi.nlm.nih.gov/bioproject/PRJNA656402) as the reference collection. We were unable to perform comparisons with the Global Microbiome Conservancy (GMbC)[60] because of restricted access. We performed a pairwise ANI alignment between CGR2 and the reference collection using fastANI (v1.32). The

clusters were considered to match if the ANI value was higher than 95%.

## Glycan-degrading CAZyme analysis

CAZymes were annotated with dbCAN (v2.0)[61]. Cellulose-, inulin- and pectin-degrading related CAZyme families were identified in the dbCAN-PUL database[62] by the link https://bcb.unl.edu/dbcan_pul/Webserver/static/DBCAN-PUL/. Preparing for the subsequent pathway analyses, KofamKOALA[63] was used to obtain enzyme information. In order to obtain more annotation, the parameter "-f detail-tsv" was set in KofamKOALA. Complete pathways of inulin-glucose, cellulose-glucose, pectin-glucose, pyruvate-acetate, pyruvate-propionate, and pyruvate-butyrate were collected by referring to KEGG pathways, map00051, map00500, map00040, map00010, map00620, map00640, map00650, and detailed pathways have been listed in Supplementary Data 5 and can be found in Supplementary Fig. 10. All enzymes of these pathways were included as filter criteria to screen for potential glycan-degrading and SCFA-producing strains. The R function ggtree was used to visualize the presence and absence of the selected CAZyme families. Vegdist and pcoa function were used in PCoA on the gene numbers of CAZyme families involved in HMO degradation.

## SMBGs identification, clustering, and network analysis

Identification of SMBGs was performed by anti-SMASH (v4.2.0)[25]. BiG-SCAPE v1.0.1[64] was used to cluster the identified SMBGs and MIBiG (version 1.4)[65] reference BGCs into CGFs (cutoff 0.3) and classes, "include_singletons" parameter was selected to include SMBGs with a distance higher than 0.3; other parameters were the default parameters. The similarity networks of SMBGs in the same class established by BiG-SCAPE were displayed by Cytoscape (v3.8.2)[35,43]. Then CORASON[64] was used to reconstruct and visualize the multi-locus phylogeny of gene clusters of interest to explore their evolutionary relationships.

## Viral sequence prediction, selection, and comparison

VirSorter (v1.0.5) was used to mine viral signals. The results of the VirSorter mining were classified into 6 categories: category 1 (sure phage), category 2 (somewhat sure phage), category 3 (not so sure phage), category 4 (sure prophage), category 5 (somewhat sure prophage), and category 6 (not so sure prophage). Predictions classified as category 1, 2, 4, or 5 were evaluated at the level of completeness with CheckV (v0.7.0), and database of checkv-db-v0.6 for reference. Sequences classified into complete, high quality (>90% completeness), or medium quality (50–90% completeness) were selected. We compared the phage genomes with MGV and GPD, using CD-HIT (v4.8.1)[66] with parameters '-c 0.95 -G 0 -aS 0.75'.

## Clustering of phages into VCs/PCs and taxonomic classification

ORFs were called using Prodigal (v2.6.3)[67]. For the resulting protein sequences, VConTACT2 (v0.9.19)[34] was used to cluster and provide taxonomic context, with the '--raw-proteins --rel-mode 'Diamond' --proteins-fp --pcs-mode MCL --vcs-mode ClusterONE --c1-bin cluster_one-1.0.jar --db 'ProkaryoticViralRefSeq88-Merged' -t 8' option.

## Phylogenetic analysis and tree construction of phages

For the CGRv phylogenetic tree, excluding the 901 provirus sequences in CGRv, the rest of the 1919 sequences were aligned by MAFFT (v7.407)[68], with the '--auto --reorder' option. The maximum-likelihood phylogenetic tree was constructed by FastTree (v2.1.3)[69]. The resulting tree was visualized using iTOL (v6.1.1)[54].

## Genomics comparison analysis

Mummer (v3.22) and lastz (v1.03.73) were used to align the assembled genomes to the reference genome (*C. aerofaciens* ATCC 25986, GCA_000169035.1) with default parameters to call the homozygous SNPs and small size insertion and deletion variants as previous described[70]. The phylogenetic tree was constructed using the SNP alignment sequences by TreeBeST (v1.9.2) with "phyml" model and bootstraps value "1000", and then all trees were visualized by using iTOL (v.6.1.1)[54]. Genomes were clustered into five groups based on the 95% ANI threshold of CDS sequences. For the gene functional annotation, BLAST (v2.2.26) was used to align the gene sequences to KEGG (v87) for annotation with E-value <1e−5.

## Statistical analysis

Statistical tests were performed using R v4.0.3. Hierarchical clustering of genomes was performed using hclust package for R with distance of 0.05. All *P* values were calculated using the Wilcoxon rank-sum test (two-sided), except for the significance analysis of microbial communities in healthy cohorts of China, HMP, and the Netherlands. The community ordination of the 527 clusters in all metagenomes was performed with the R functions vegdist (Bray–Curtis dissimilarities) and cmdscale (Principal Co-ordinates Analysis, PCoA). R function envfit was used to test the correlation between either categorical data or continuous variables with the coordinates of microbial communities and the significance was assessed using permutation.

## Reporting summary

Further information on research design is available in the Nature Portfolio Reporting Summary linked to this article.

## Data availability

The genome data generated in this study have been deposited into CNSA[55] of CNGBdb[56] with accession number CNP0000126 and CNP0001833, and NCBI under the projects PRJNA482748 and PRJNA903559. All the bacterial strains in CGR2 have been deposited in China National GeneBank (CNGB), a non-profit, public-service-oriented organization in China. The strain information, including taxonomy, donor, and culture conditions can be found and accessed through https://db.cngb.org/codeplot/datasets/CGR2. The databases used in this study include GTDB database Release 07-RS207 (https://data.gtdb.ecogenomic.org/releases/release207/), dbCAN-PUL (https://bcb.unl.edu/dbcan_pul/Webserver/static/DBCAN-PUL/), KEGG database (https://www.genome.jp/kegg), MIBiG version 1.4 (https://mibig.secondarymetabolites.org/), checkv-db-v0.6 (https://portal.nersc.gov/CheckV/), and Genebank (www.ncbi.nlm.nih.gov/genbank/). The datasets used in this study include human gut metagenome sequencing data of a Chinese cohort (a part of 4D-SZ) (https://db.cngb.org/search/project/CNP0000426/), HMP (https://portal.hmpdacc.org/), the Netherlands cohort (https://ega-archive.org/studies/EGAS00001005027), UHGG (http://ftp.ebi.ac.uk/pub/databases/metagenomics/mgnify_genomes), BIO-ML (https://www.ncbi.nlm.nih.gov/bioproject/PRJNA544527), hGMB (https://www.ncbi.nlm.nih.gov/bioproject/PRJNA656402), MGV (https://portal.nersc.gov/MGV/), and GPD (https://ftp.ebi.ac.uk/pub/databases/metagenomics/genome_sets/gut_phage_database/).

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

## Acknowledgements

This work was supported by grants from National Key Research and Development Program of China (No. 2018YFC1313801), National Natural Science Foundation of China (No. 32100009), and Natural Science Foundation of Guangdong Province, China (No. 2019B020230001). The data analysis was supported by the Henan Supercomputer Center. We also thank the colleagues at BGI-Shenzhen for sample collection, and discussions, and China National GeneBank (CNGB) Shenzhen for DNA extraction, library construction, and sequencing.

## Author contributions

Conceived and designed the experiments: Y.Z. and L.X. Performed the experiments: Y.Z., X.L., and X.Jing. Analyzed the data: Y.Z., X.L., L.X., T.H., J.C., H.L., Z.W., and J.Z. Contributed reagents/materials/analysis tools: Y.Z., C.Y., X.Jin., X.X., W.Z., T.Y., H.Y., and J.W. Wrote the manuscript: Y.Z., X.L., L.X., T.H., J.C., H.L., K.K., and J.Z. Revised the manuscript: K.K. All authors commented on the manuscript.

## Competing interests

The authors declare no competing interests.
