## [Peer Review File · Nature Communications]

REVIEWER COMMENTS

Reviewer #1 (Remarks to the Author):

In this manuscript, Zou et al. provide an expanded version of their previously released bacteria culture collection referred as the Cultivated Genome Reference (CGR). Culturing efforts such as these are always important for the microbiome community as they provide the means to mechanistically test the biological functions of the human gut microbiome. This work represents one of the largest such efforts in the field, resulting from the culturing of over 20,000 isolates. The authors also do a good job in providing some additional analyses in relation to their previous work (namely related to phage-bacteria relationships and investigating the strain diversity of *Collinsella aerofaciens*). I have a few comments/suggestions that I believe would provide even greater value to the study.

1) The authors used GTDB-Tk v1.3 for their analysis, but the most recent version (v2.1) received a massive overhaul and expansion, more than doubling the number of species (~30,000 in v1.3 to >60,000 in v2.1). I suggest the authors at least check and comment in the manuscript how many of their species/genera are indeed novel in relation to the most up-to-date version of GTDB.

2) Why was 10% used as the contamination threshold? Although the risk of contamination is lower with isolate genomes, it is clear from Supp Fig 1 that many of the genomes recovered were highly contaminated (some with as high as 600% contamination, i.e., 6/7 genomes in one). I suggest the authors to use a stricter 5% threshold to avoid contaminating public databases.

3) Given the massive recent efforts in expanding our understanding of the human gut bacteriophage diversity (Camarillo et al. Cell 2021 and Nayfach et al. Nature Micro 2021), it is important that the authors compare the phages they recovered to these existing collections.

4) A bit more discussion/analysis on the novel species/genera is warranted. How abundant/prevalent are these species? Are they exclusive/overrepresented in the Chinese population? Do they provide any meaningful improvements in metagenomic read mapping classification in Asian and non-Asian populations?

5) The analysis focused on *Collinsella aerofaciens* is very interesting. The authors state that these genomes have a higher rate of variability within intergenic regions. Did the authors check if there are

specific genome locations with increased variability (i.e., mutational hotspots) that could point to some level of adaptive evolution to the human gut?

6) Strain availability: The authors mention that the bacterial strains were deposited in the China National GeneBank. However, no further information is provided on how to find and access these samples. Any identifiers/accessions related to their submission should be provided so users can easily identify and access this valuable resource.

7) Lines 200-202. The interpretation that Bacteroidota represents a unique cluster because of “less loss or gain” of CAZy genes seems a bit bizarre. Couldn’t this distinct cluster just be a result of Bacteroidota having its own unique CAZy composition? Not necessarily that they have more or less CAZy genes.

8) Line 490: What was the authors definition of “healthy donors”. Important to clearly state the criteria for this.

9) There are a number of typographical errors throughout the manuscript that should be checked. I have listed below a few:

Line 173: “most strain in CGR2 had potentially” -> “most strains in CGR2 had the potential”

Line 182: “play important role” -> “play important roles”

Line 498: “Gene number” -> “Gene numbers”

Reviewer #2 (Remarks to the Author):

In this manuscript, "The genomic landscape of reference genomes of cultivated human gut bacteria," the authors created an extensive isolates collection with accompanying whole genomes. This isolate collection is the second phase of a previous work published in Nature Biotechnology (CGR). By culturing over 20,000 isolates and sequencing the whole genome of >3000 isolates, the authors claimed that their new isolate collection contains a substantial level of novel species. The authors

then analyzed CaZyme, Biosynthetic gene clusters (BGCs) to show unexplored functional diversity and prophage-host associations. Lastly, the authors dig deep into *Collinsella aerofaciens*, a prevalent yet under-characterized gut species, and demonstrate the intra-species diversity of this species.

The resource generated by this work will be of great interest to microbiome researchers. While the analysis in this paper provides an overview of the impact of the dataset, several sections of the manuscript suffer from a lack of rigorous method.

Major:

1. One of the most significant results is that the authors claimed to discover 222 novel species. This could be a potentially very impactful contribution to the microbiome field. However, the way that the authors define "novel species" is not very convincing, as it seems that the authors only compare species-level genome clusters to the GTDB database and label clusters without a species-level annotation as "novel." This is not a widely acknowledged method, and more evidence/analysis is required to support this claim. Are these species ever discovered in the several large MAG databases or UHGG? If so, some species should be "species that have not been cultured before" rather than "novel species." Or did the author actually just intend to say "species without a proper name?" If there really are many "novel species" that have never been reported elsewhere, the author should describe a few examples to illustrate some features of these novel species. In addition, authors might consider replace "novel" with a more accurate definition.

2. While results from the CaZyme and BGC sections seem to yield important insights, it is unclear to me whether the CGR2 is uniquely valuable or other datasets will also yield similar results.

Specific comments:

Line 70: MAGs can also provide information for culturable species. Better to replace it with "culture-independent."

Line 87: "The CGR2 comprises 54 novel species with high-quality genomes that match 100 MAGs without culture isolates." Unclear what this sentence mean.

Line 99: how are the isolates selected for WGS? Is there any information for the isolates without WGS? If not, what's the value of reporting the remaining ~16,000 isolates?

Line 99-101: how many isolates are from the previous publication of CGR? This information is not evident throughout the abstract and main text. I realize that there are ~1800 new whole genomes only when reading the figure 1 legend, and ~1500 genomes are from CGR.

Line 109-110: as mentioned above, the claim that 222 species are "novel" is ambiguous and requires more analysis.

Line 116: authors need to provide some information about the "BGI cohort." E.g., do the BGI cohort and the CGR cohort share human donors?

Line 116: "taxa" is a vague term to describe results. Did the author mean species, genus, or something else?

Line 115-120. The comparison between CGR2 and BGI is problematic. The taxonomical composition of BGI metagenomes is inferred with MetaPhlan2. It is not guaranteed that the nomenclature from MetaPhlan2 and GTDB is consistent. The same species (or genus) might have different names between MetaPhlan2 and GTDB. In my view, comparing metagenome species composition with isolate collections is challenging. To detect whether isolates-associated species exist in metagenomes, the most rigorous analysis is to directly align metagenomic reads to the isolate genome and examine read distribution across the isolate genome. However, asking the question that "what are the species that present in metagenomes but not in the isolate collection" is extremely difficult, as the current metagenomic tools, such as MetaPhlan or Kraken, due to the nature of their algorithms, will likely overestimate the number of species from a microbiome.

Line 137-140: it's not clear from figure 1d that Butyricicoccaceae is the lowest family, as there is only one dot for this family. Also, what does each dot represent in figure 1d? Species?

Line 144: it's confusing to read "286 genomes mapped to 89 genomes"... Does the "286 CGR2 genomes" mean 286 isolates?

Line 280: it seems quite exciting and surprising that 4 VCs are associating with multiple phyla. I'd like to see more details of these VCs. For example, how different are the sequences of the same VC differ between bacterial phyla? Did authors check sequencing depth over these VC regions to rule out contamination during library prep?

Line 308-301. It's unclear to me how the authors concluded that "high genomic diversity is likely related to differences in non-protein-coding intergenic regions."

Reviewer #3 (Remarks to the Author):

Congratulation on this tremendous amount of work with cultured human gut bacteria.

This is an incremental work, i.e. a previous version was published previously. The authors compare their data to other resources, but I think this part of the work is incomplete. The authors are invited to perform new analyses and create a figure solely dedicated to showing the added value of the second version of this collection when compared to the first one and also to ALL, UP-TO-DATE existing collections of isolates from others. The novelty in this collection is currently over-emphasized. For instance the work by Groussin et al. must be considered instead of Poyet et al. 2019. If the authors have trouble accessing the data, this could be acknowledged in the manuscript, as this is also another major concern in this study!

One major incentive of the work is to release a collection of isolates and genomes to help the research community. There are major concerns about the availability of strains and their genomes. Using the information provided by the authors, accession CNP0000126 returned 4,563 entries (most likely the first version of the collection?) and CNP00001833 zero. Hence the new genomes that justifies the work are not accessible.

Moreover, information about the collection is cryptic. Exploring the website of the China National GeneBank, I was able to find 1,894 isolates from the intestine, which is far away from the 3,324 isolates claimed by the authors. A clear link to the resource must be given in the paper. There is absolutely no proof of deposition provided currently as well as information on public accessibility of the isolates or any restrictions of use linked to them. This is not acceptable in the era of FAIR data.

I feel the data should be compared to cohorts and their metagenomes other than BGI to be more representative

In figure 1, I am concerned by the presence of Planctomytetota and other non-gut specific microbes. It seems this is related to the sequencing data as no isolates obtained, emphasizing the concern about contaminations in the sequencing data. The analysis seems biased. Please clarify and modify.

REVIEWER COMMENTS

Reviewer #1 (Remarks to the Author):

In this manuscript, Zou et al. provide an expanded version of their previously released bacteria culture collection referred as the Cultivated Genome Reference (CGR). Culturing efforts such as these are always important for the microbiome community as they provide the means to mechanistically test the biological functions of the human gut microbiome. This work represents one of the largest such efforts in the field, resulting from the culturing of over 20,000 isolates. The authors also do a good job in providing some additional analyses in relation to their previous work (namely related to phage-bacteria relationships and investigating the strain diversity of *Collinsella aerofaciens*). I have a few comments/suggestions that I believe would provide even greater value to the study.

1) The authors used GTDB-Tk v1.3 for their analysis, but the most recent version (v2.1) received a massive overhaul and expansion, more than doubling the number of species (~30,000 in v1.3 to >60,000 in v2.1). I suggest the authors at least check and comment in the manuscript how many of their species/genera are indeed novel in relation to the most up-to-date version of GTDB.

Response:

We thank the reviewer for this suggestion. We have re-annotated the taxonomy using GTDB-Tk v2.1.0 and the latest version of the reference database (r207). The result showed that CGR2 presents 179 unknown species (potentially new species) and 21 potentially new genera (Line 108). We have updated the taxonomy information throughout the manuscript.

2) Why was 10% used as the contamination threshold? Although the risk of contamination is lower with isolate genomes, it is clear from Supp Fig 1 that many of the genomes recovered were highly contaminated (some with as high as 600% contamination, i.e., 6/7 genomes in one). I suggest the authors to use a stricter 5% threshold to avoid contaminating public databases.

Response:

We concur with the reviewer's opinion that a strict contamination threshold is important. After checking the contamination of our genomes, we point out that the contamination of 3,323 genomes is < 5%, that is, only one genome has a contamination rate of 5.27%. This genome had no influence on the results of the full text.

3) Given the massive recent efforts in expanding our understanding of the human gut bacteriophage diversity (Camarillo et al. Cell 2021 and Nayfach et al. Nature Micro 2021), it is important that the authors compare the phages they recovered to these existing collections.

Response:

We compared CGRv with GPD and MGv (Camarillo et al. Cell 2021 and Nayfach et al. Nature Micro 2021) according to the reviewer's suggestion. 564 phage sequences from GPD, and 556 sequences from MGv matched the phage genomes of CGRv, and 1,707 sequences from CGRv were still novel at the vOTU level (Line 278-280 and **Supplementary Fig. 15b**).

4) A bit more discussion/analysis on the novel species/genera is warranted. How abundant/prevalent

are these species? Are they exclusive/overrepresented in the Chinese population? Do they provide any meaningful improvements in metagenomic reads mapping classification in Asian and non-Asian populations?

Response:

We have added an investigation of the abundance/prevalence of these potentially novel species/genera in the metagenomes of three cohorts from China, HMP, and the Netherlands. We found that the average abundance of these 179 potentially new species (defined as “unknown species” in the updated manuscript) in the Chinese population was 0.08%, which was significantly higher than that in the other two cohorts (**Supplementary Fig. 4a-b**). Based on the genomes of the CGR2, we found that the reads mapping classification was significantly improved for both Chinese and non-Chinese metagenomes by adding the unknown species (**Supplementary Fig. 4c**, Line 128-136).

5) The analysis focused on *Collinsella aerofaciens* is very interesting. The authors state that these genomes have a higher rate of variability within intergenic regions. Did the authors check if there are specific genome locations with increased variability (i.e., mutational hotspots) that could point to some level of adaptive evolution to the human gut?

Response:

We checked the 10 CDSs with top mutation frequency and found that the variant type and frequency appeared to correlate with cluster and country (**Supplementary Fig. 17c**). However, since we lack more phenotypic data, it cannot be determined whether these variants would point to adaptive evolution.

6) Strain availability: The authors mention that the bacterial strains were deposited in the China National GeneBank. However, no further information is provided on how to find and access these samples. Any identifiers/accessions related to their submission should be provided so users can easily identify and access this valuable resource.

Response:

Strain information, including taxonomic information and culture conditions, can be found and accessed through this website, <https://db.cngb.org/codeplot/datasets/CGR2>. Instructions for access to this resource have been added to DATA AVAILABILITY (Line 834-835).

7) Lines 200-202. The interpretation that Bacteroidota represents a unique cluster because of “less loss or gain” of CAZy genes seems a bit bizarre. Couldn't this distinct cluster just be a result of Bacteroidota having its own unique CAZy composition? Not necessarily that they have more or less CAZy genes.

Response:

We concur with the reviewer's comments and we have deleted the sentence about “loss and gain”.

8) Line 490: What was the authors definition of “healthy donors”. Important to clearly state the criteria for this.

Response:

The ones reported without any diagnosed disease were considered healthy donors. We have stated the criteria in the Methods (Line 511-513).

9) There are a number of typographical errors throughout the manuscript that should be checked. I have listed below a few:

Line 173: "most strain in CGR2 had potentially" -> "most strains in CGR2 had the potential"

Response: We have corrected the sentence accordingly (Line 191).

Line 182: "play important role" -> "play important roles"

Response: We have corrected "role" to "roles" (Line 200).

Line 498: "Gene number" -> "Gene numbers"

Response: We have corrected "Gene number" to "Gene numbers" (Line 527).

Reviewer #2 (Remarks to the Author):

In this manuscript, "The genomic landscape of reference genomes of cultivated human gut bacteria," the authors created an extensive isolates collection with accompanying whole genomes. This isolate collection is the second phase of a previous work published in Nature Biotechnology (CGR). By culturing over 20,000 isolates and sequencing the whole genome of >3000 isolates, the authors claimed that their new isolate collection contains a substantial level of novel species. The authors then analyzed CaZyme, Biosynthetic gene clusters (BGCs) to show unexplored functional diversity and prophage-host associations. Lastly, the authors dig deep into *Collinsella aerofaciens*, a prevalent yet under-characterized gut species, and demonstrate the intra-species diversity of this species.

The resource generated by this work will be of great interest to microbiome researchers. While the analysis in this paper provides an overview of the impact of the dataset, several sections of the manuscript suffer from a lack of rigorous method.

Major:

1. One of the most significant results is that the authors claimed to discover 222 novel species. This could be a potentially very impactful contribution to the microbiome field. However, the way that the authors define "novel species" is not very convincing, as it seems that the authors only compare species-level genome clusters to the GTDB database and label clusters without a species-level annotation as "novel." This is not a widely acknowledged method, and more evidence/analysis is required to support this claim. Are these species ever discovered in the several large MAG databases or UHGG? If so, some species should be "species that have not been cultured before" rather than "novel species." Or did the author actually just intend to say "species without a proper name?" If there really are many "novel species" that have never been reported elsewhere, the author should describe a few examples to illustrate some features of these novel species. In addition, authors might consider replace "novel" with a more accurate definition.

Response:

We originally defined potentially novel species as clusters without a proper species annotation in the database based on sequence similarity. We find that some of these "potentially new species" were present in other databases, such as UHGG, BIO-ML and hGMB, but they were all unknown species. We suggest that these bacteria should be defined as "unknown species" which would be

more accurate.

Furthermore, we found that half of these “unknown species” were still not represented in UHGG, BIO-ML and hGMB, including one cluster being annotated only at the class-level (**Supplementary Fig. 8a and Supplementary Table 2**). It is worth noting that these underrepresented novel clusters were widely distributed in the metagenomes from China, the HMP, and the Netherlands (**Supplementary Fig. 8b-d**).

In addition, there are still 31 “unknown species” that are only represented by MAGs, while we provide the first cultured strains to facilitate subsequent taxonomic characterization (**Supplementary Fig. 8a**).

2. While results from the CaZyme and BGC sections seem to yield important insights, it is unclear to me whether the CGR2 is uniquely valuable or other datasets will also yield similar results.

Response:

Kaoutari *et al.* 2013 have described gut bacteria from the Bacteroidota phylum encoding more CAZymes, and our study came to similar conclusions. We used the large scaled culture-based genomes to conduct the CaZyme analysis furthering the understanding of the potential roles of gut bacteria in the metabolism of carbohydrates (pectin, cellulose, and inulin).

For BGC, Donia *et al.* 2014 focused on the secondary metabolite resources of bacterial strains isolated from humans and showed that there are 7 types of BGCs present in the human microbiota. Almeida *et al.* 2019 screened BGCs from MAGs of the human gut, and showed that more than 70% were new BGCs. In our study, a total of 24 BGC types were obtained through large-scale mining of human gut bacterial strains, and 99.89% of the BGCs were novel, showing that gut microbes are a rich source of diverse secondary metabolites. In addition, mining efforts in cultured genomes can provide accurate information and facilitate subsequent experimental validation.

Ref:

1. El Kaoutari, A., Armougom, F., Gordon, J. I., Raoult, D. & Henrissat, B. The abundance and variety of carbohydrate-active enzymes in the human gut microbiota. *Nat Rev Microbiol* **11**, 497-504, doi:10.1038/nrmicro3050 (2013).
2. Donia, M. S. *et al.* A systematic analysis of biosynthetic gene clusters in the human microbiome reveals a common family of antibiotics. *Cell* **158**, 1402-1414, doi:10.1016/j.cell.2014.08.032 (2014).
3. Almeida, A. *et al.* A new genomic blueprint of the human gut microbiota. *Nature* **568**, 499-504 (2019).

Specific comments:

Line 70: MAGs can also provide information for culturable species. Better to replace it with "culture-independent."

Response:

We have changed the wording according to the reviewer's suggestion (Line 69).

Line 87: "The CGR2 comprises 54 novel species with high-quality genomes that match 100 MAGs without culture isolates." Unclear what this sentence mean.

Response:

We have rewritten the sentence to make it clearer (Line 87).

Line 99: how are the isolates selected for WGS? Is there any information for the isolates without WGS? If not, what's the value of reporting the remaining ~16,000 isolates?

Response:

The 16S rRNA genes of each isolate were sequenced and annotated by EZBioCloud. All the isolates were grouped into species-level clusters based on a threshold of 98.7% identity of the 16S rRNA gene sequence. Our strategy for selecting strains for WGS were (1) representation of candidates for new taxa, (2) covering as many taxa as possible of the strains cultivated in the study, (3) important species from various donors. (Line 514-Line 517, Line 522-Line 525).

Except for the 3,324 bacteria for which genomes are available, the rest of the bacterial strains were redundant with the 3,324 bacteria at the species level. These bacteria can be sequenced subsequently and used for pan genome and comparative genomic studies in the future.

Line 99-101: how many isolates are from the previous publication of CGR? This information is not evident throughout the abstract and main text. I realize that there are ~1800 new whole genomes only when reading the figure 1 legend, and ~1500 genomes are from CGR.

Response:

1,520 isolates are from the CGR. We have added this information to Results. (Line 104)

Line 109-110: as mentioned above, the claim that 222 species are "novel" is ambiguous and requires more analysis.

Response:

Considering the reviewer's suggestion, we have corrected "novel species" to "unknown species" and performed more analysis (Line 110, Line 128-136, and Line 164-172).

Line 116: authors need to provide some information about the "BGI cohort." E.g., do the BGI cohort and the CGR cohort share human donors?

Response:

The "BGI cohort" is from a previously published metagenomic study of a Chinese population (Ref.1). In order to be consistent with previously published articles, we changed the name to "4D-SZ". Our study is an independent piece of work, and no donors are shared with 4D-SZ.

Ref1: Jie, Z. *et al.* A transomic cohort as a reference point for promoting a healthy human gut microbiome. *Medicine in Microecology* **8**, doi:10.1016/j.medmic.2021.100039 (2021).

Line 116: "taxa" is a vague term to describe results. Did the author mean species, genus, or something else?

Response:

The word "taxa" is incorrect here, we have replaced it with "species" in the updated analysis.

Line 115-120. The comparison between CGR2 and BGI is problematic. The taxonomical composition of BGI metagenomes is inferred with MetaPhlan2. It is not guaranteed that the

nomenclature from Metaphlan2 and GTDB is consistent. The same species (or genus) might have different names between Metaphlan2 and GTDB. In my view, comparing metagenome species composition with isolate collections is challenging. To detect whether isolates-associated species exist in metagenomes, the most rigorous analysis is to directly align metagenomic reads to the isolate genome and examine read distribution across the isolate genome. However, asking the question that "what are the species that present in metagenomes but not in the isolate collection" is extremely difficult, as the current metagenomic tools, such as MetaPhlan or Kraken, due to the nature of their algorithms, will likely overestimate the number of species from a microbiome.

Response:

Thank you for pointing this out. According to your suggestion, we have aligned the genomes of our collection to metagenomic reads of different cohorts. Details can be found in the updated manuscript (Line 115-127).

Line 137-140: it's not clear from figure 1d that Butyricicoccaceae is the lowest family, as there is only one dot for this family. Also, what does each dot represent in figure 1d? Species?

Response:

Thank you for pointing out this problem in our manuscript. We have re-written this part (Line 147). For **figure 1d**, a dot represents a UHGG genome, and differently colored dots indicate different families. This detailed information has been to the legend (Line 458-459).

Line 144: it's confusing to read "286 genomes mapped to 89 genomes"... Does the "286 CGR2 genomes" mean 286 isolates?

Response:

Yes, "286 CGR2 genomes" mean 286 isolates. We have revised the sentence to make it correct (Line 152-Line 154).

Line 280: it seems quite exciting and surprising that 4 VCs are associating with multiple phyla. I'd like to see more details of these VCs. For example, how different are the sequences of the same VC differ between bacterial phyla? Did authors check sequencing depth over these VC regions to rule out contamination during library prep?

Response:

According the reviewer's suggestion, we conducted more analyses of these VCs associated with multiple phyla. We found no significant difference between these 4 phages at the gene and protein level. On the other hand, we have checked the sequencing depth over these 4 VC regions, and found the coverage of sequencing depth was uniform distributed across these VC regions implying that contamination would not be likely.

Line 308-301. It's unclear to me how the authors concluded that "high genomic diversity is likely related to differences in non-protein-coding intergenic regions."

Response:

Based on the ANI value (95% threshold), the *C. aerofaciens* genomes were divided into 19 clades, while based on the SNPs in the CDS region *C. aerofaciens* genomes could be divided into 5 groups. This indicated that *C. aerofaciens* has greater diversity at the genome-wide level than in CDS regions, so mutations in non-protein-coding intergenic regions is one cause of high genomic

diversity. We have re-phrased this part (Line 326-327).

Reviewer #3 (Remarks to the Author):

Congratulation on this tremendous amount of work with cultured human gut bacteria.

This is an incremental work, i.e. a previous version was published previously. The authors compare their data to other resources, but I think this part of the work is incomplete. The authors are invited to perform new analyses and create a figure solely dedicated to showing the added value of the second version of this collection when compared to the first one and also to ALL, UP-TO-DATE existing collections of isolates from others. The novelty in this collection is currently over-emphasized. For instance the work by Groussin et al. must be considered instead of Poyet et al. 2019. If the authors have trouble accessing the data, this could be acknowledged in the manuscript, as this is also another major concern in this study!

Response:

Thank you for your comments. We have updated the comparisons of CGR and its second version CGR2 to existing collections at **Supplementary Fig. 7b**. The result shows that compared with the existing collection, CGR has 144 unique clusters, while CGR2 contributes additional 45 clusters. For the selection of collections for comparison, we noticed the work of GMbC conducted by Groussin et al. but have had trouble accessing the data. We have stated this in the Methods.

One major incentive of the work is to release a collection of isolates and genomes to help the research community. There are major concerns about the availability of strains and their genomes. Using the information provided by the authors, accession CNP0000126 returned 4,563 entries (most likely the first version of the collection?) and CNP00001833 zero. Hence the new genomes that justifies the work are not accessible.

Response:

The genomes are currently publicly accessible with accession number CNP0000126 and CNP0001833. In addition, the genomes can also be accessed in NCBI under the projects PRJNA482748 and PRJNA903559.

Moreover, information about the collection is cryptic. Exploring the website of the China National GeneBank, I was able to find 1,894 isolates from the intestine, which is far away from the 3,324 isolates claimed by the authors. A clear link to the resource must be given in the paper. There is absolutely no proof of deposition provided currently as well as information on public accessibility of the isolates or any restrictions of use linked to them. This is not acceptable in the era of FAIR data.

Response:

Strain information, including taxonomic information and culture conditions, can be found and accessed through this website, <https://db.cngb.org/codeplot/datasets/CGR2>. Instructions for this resource have been added to DATA AVAILABILITY (Line 837-837). Based on our original CGR, published in Nature Biotechnology, we have provided more than 50 bacterial strains to several researchers.

I feel the data should be compared to cohorts and their metagenomes other than BGI to be more representative

In figure 1, I am concerned by the presence of Planctomycetota and other non-gut specific microbes. It seems this is related to the sequencing data as no isolates obtained, emphasizing the concern about contaminations in the sequencing data. The analysis seems biased. Please clarify and modify.

Response:

Thanks for the suggestion. We have aligned metagenomic reads from healthy cohorts from China, the Netherlands, and HMP to our representative genomes, and analyzed the distribution of genomes in different metagenome cohorts; the result has been updated in line 115-127 and **Figure 1b**.

As for Planctomycetota, it has recently been linked to human pathology as an opportunistic pathogen. doi: 10.3389/fcimb.2020.519301.

REVIEWERS' COMMENTS

Reviewer #1 (Remarks to the Author):

I thank the authors for addressing all my comments and I have no further concerns.

Reviewer #3 (Remarks to the Author):

Thanks for clarifying the major points about the accessibility of the isolates and their data/metadata.

1/ When navigating the CGR2 website using the link provided by the authors, the new pages that opened after clicking on some of the links were available only in Chinese. This may have to be solved for easy navigation by international readers.

2/ It will be important for easy reuse of this resource that all genomes are available at once. Whilst important that all genomes are available individually (current state), I believe that a repository such as zenodo or whatever the authors may prefer will be helpful for easy access all-at-a-time. Sorry if I miss the current way the authors are dealing with this.

3/ L52: delete „remarkable“

4/ L79: “Culturomics, a culturing approach employing multiple culture conditions”. And any other places in the manuscript referring to culturomics. Anaerobic culture using multiple conditions was already done in the 1960s. I strongly recommend that the term "culturomics" is removed from the manuscript. The nice studies by the authors do not need it.

5/ Bacterial phyla have been reclassified and validly named in October 2021. I believe the authors should revise their manuscript, so it contains the updated nomenclature.

6/ Similarly, the genus *Prevotella* was recently reclassified and split into multiple genera; I believe this is unfair at this stage to ask the authors to rename all their isolates considering that the work aforementioned and associated validation of name is relatively recent (links below). nevertheless, considering the importance of “*Prevotella*” spp. in the human gut and thus in this manuscript, the authors are invited to drop a note on its reclassification.

<https://pubmed.ncbi.nlm.nih.gov/36067550/>

<https://doi.org/10.1099/ijsem.0.005709>

7/ *Bacteroides vulgatus* has been reclassified as a member of the genus *Phocaeciol*; whilst this was correct in one of the supplementary table, I believe the older names appears in a few instances in the manuscript.

8/ I am not aware of *Acetatifactor intestinalis* being a valid name.

REVIEWERS' COMMENTS

Reviewer #1 (Remarks to the Author):

I thank the authors for addressing all my comments and I have no further concerns.

Response:

We would like to thank the reviewer again for taking the time to review our manuscript.

Reviewer #3 (Remarks to the Author):

Thanks for clarifying the major points about the accessibility of the isolates and their data/metadata.

1/ When navigating the CGR2 website using the link provided by the authors, the new pages that opened after clicking on some of the links were available only in Chinese. This may have to be solved for easy navigation by international readers.

Response:

We have checked the website and the link related to our CGR2 project and confirm that the link can be switched to English through the upper right corner.

2/ It will be important for easy reuse of this resource that all genomes are available at once. Whilst important that all genomes are available individually (current state), I believe that a repository such as zenodo or whatever the authors may prefer will be helpful for easy access all-at-a-time. Sorry if I miss the current way the authors are dealing with this.

Response:

We thank the reviewer for this comment. We think that it may be possible to download all genomes at the same time through a FTP server at NCBI. In addition, the “assembly” page of bioproject provides another possibility of one-click download by clicking “Download Assemblies”.

3/ L52: delete “remarkable”

Response:

We have deleted “remarkable” (Line 52).

4/ L79: “Culturomics, a culturing approach employing multiple culture conditions”. And any other places in the manuscript referring to culturomics. Anaerobic culture using multiple conditions was already done in the 1960s. I strongly recommend that the term "culturomics" is removed from the manuscript. The nice studies by the authors do not need it.

Response:

We have removed the term “culturomics”.

5/ Bacterial phyla have been reclassified and validly named in October 2021. I believe the authors should revise their manuscript, so it contains the updated nomenclature.

Response:

Thank you for pointing this out. We have checked the bacterial phyla in the manuscript, figures, and tables, and changed them to the correct and valid published names.

6/ Similarly, the genus *Prevotella* was recently reclassified and split into multiple genera; I believe this is unfair at this stage to ask the authors to rename all their isolates considering that the work aforementioned and associated validation of name is relatively recent (links below). nevertheless, considering the importance of “*Prevotella*” spp. in the human gut and thus in this manuscript, the authors are invited to drop a note on its reclassification.

<https://pubmed.ncbi.nlm.nih.gov/36067550/>

<https://doi.org/10.1099/ijsem.0.005709>

Response:

We have made a statement about the reclassification of *Prevotella* in the manuscript. (Lines 124-127)

7/ *Bacteroides vulgatus* has been reclassified as a member of the genus *Phocaeciol*; whilst this was correct in one of the supplementary table, I believe the older names appears in a few instances in the manuscript.

Response:

We have corrected “*Bacteroides vulgatus*” to “*Phocaeicola vulgatus*” (Line 299, and supplementary figure 16).

8/ I am not aware of *Acetatifactor intestinalis* being a valid name.

Response:

We have corrected “*Acetatifactor intestinalis*” to “*Waltera intestinalis*”, the valid name of this species (Line 259, figure 2, figure 3, supplementary data 3, supplementary data 6, and supplementary data 7).